# Development and External Validation of Partial Proportional Odds Risk Prediction Models for Cancer Stage at Diagnosis among Males and Females in Canada

**DOI:** 10.3390/cancers15143545

**Published:** 2023-07-08

**Authors:** Timofei Biziaev, Michelle L. Aktary, Qinggang Wang, Thierry Chekouo, Parveen Bhatti, Lorraine Shack, Paula J. Robson, Karen A. Kopciuk

**Affiliations:** 1Department of Mathematics and Statistics, University of Calgary, Calgary, AB T2N 4N2, Canada; 2Faculty of Kinesiology, University of Calgary, Calgary, AB T2N 1N4, Canada; 3Cancer Epidemiology and Prevention Research, Cancer Care Alberta, Alberta Health Services, Calgary, AB T2S 3C3, Canada; 4Division of Biostatistics, School of Public Health, University of Minnesota, Minneapolis, MN 55455, USA; 5Cancer Control Research, BC Cancer, Vancouver, BC V5Z 1L3, Canada; 6School of Population and Public Health, University of British Columbia, Vancouver, BC V6T 1Z3, Canada; 7Cancer Surveillance and Reporting, Alberta Health Services, Calgary, AB T2S 3C3, Canada; 8Department of Agricultural, Food and Nutritional Science and School of Public Health, University of Alberta, Edmonton, AB T6G 2P5, Canada; 9Cancer Care Alberta and Cancer Strategic Clinical Network, Alberta Health Services, Edmonton, AB T5J 3H1, Canada; 10Departments of Oncology, Community Health Sciences, University of Calgary, Calgary, AB T2N 4N2, Canada

**Keywords:** Alberta’s Tomorrow Project, British Columbia Generations Project, calibration, discrimination, external validation, cancer stage at diagnosis

## Abstract

**Simple Summary:**

Diagnosing most cancers at an earlier stage improves outcomes because treatments are more effective and often less invasive. This study looked at the health patterns of adults enrolled in Alberta’s Tomorrow Project before they were diagnosed with cancer to identify factors related to cancers that are caught early or late. These identified factors were then tested in a similar group of adults from the British Columbia Generations Project who also subsequently developed cancer, to see how well they could predict stage at diagnosis. The confirmed health patterns can be used to improve cancer prevention programs and activities to find cancer earlier in people who are at most risk of being diagnosed with late-stage cancer.

**Abstract:**

Risk prediction models for cancer stage at diagnosis may identify individuals at higher risk of late-stage cancer diagnoses. Partial proportional odds risk prediction models for cancer stage at diagnosis for males and females were developed using data from Alberta’s Tomorrow Project (ATP). Prediction models were validated on the British Columbia Generations Project (BCGP) cohort using discrimination and calibration measures. Among ATP males, older age at diagnosis was associated with an earlier stage at diagnosis, while full- or part-time employment, prostate-specific antigen testing, and former/current smoking were associated with a later stage at diagnosis. Among ATP females, mammogram and sigmoidoscopy or colonoscopy were associated with an earlier stage at diagnosis, while older age at diagnosis, number of pregnancies, and hysterectomy were associated with a later stage at diagnosis. On external validation, discrimination results were poor for both males and females while calibration results indicated that the models did not over- or under-fit to derivation data or over- or under-predict risk. Multiple factors associated with cancer stage at diagnosis were identified among ATP participants. While the prediction model calibration was acceptable, discrimination was poor when applied to BCGP data. Updating our models with additional predictors may help improve predictive performance.

## 1. Introduction

Cancer is the leading cause of morbidity and mortality in Canada, with an estimated two in five Canadians expected to develop cancer in their lifetime and one in four Canadians expected to die from the disease [1]. Cancer incidence and mortality rates are slightly higher in males than females [1,2]. These differences may be partly attributable to the different types of cancers developed among males and females and to differences in biology, risk factors, and health practices [1,2].

Cancer stage at diagnosis is an important determinant of cancer survival [3]. Individuals diagnosed at a later stage generally have fewer treatment options and lower overall five-year net survival rates [3]. For instance, in Canada, the 5-year survival rate for late-stage (i.e., stage IV) colon cancer is only 11%, compared to 92% for early stage (i.e., stage I) [3]. Therefore, diagnosing and treating cancer at an earlier stage is widely recognized as a key priority [4,5].

Cancer screening programs can help identify cancer in its early stages, thus enabling curative treatment and enhancing survival [6]. However, organized screening programs in Canada exist for only some cancers (i.e., colorectal, cervical, and breast cancers) [6], and despite the availability of these programs, advanced cancer diagnoses persist [3]. Previous studies have identified several factors associated with cancer stage at diagnosis, such as age, smoking status, and socioeconomic position [7,8,9,10,11,12,13,14,15]. In addition, cancer stage at diagnosis has been found to differentially affect males and females [7,8,16] and, thus, factors associated with cancer stage at diagnosis may also differ by sex [8,15]. For instance, we previously identified sex-specific factors, such as social support for males and reproductive factors for females, associated with colorectal cancer among adults living in Alberta, Canada [8].

Risk prediction models are increasingly used to support clinical decision-making [17]. While a number of cancer risk prediction models have been developed, most involve predicting dichotomous outcomes (e.g., presence versus absence of cancer) [18,19,20,21,22]. Fewer risk prediction models have been developed to predict ordinal outcomes, such as cancer stage at diagnosis [23,24,25,26,27,28]. Identifying factors predictive of cancer stage at diagnosis is relevant and clinically important given that stage at diagnosis generally determines cancer treatment options and their effectiveness, and overall cancer survival [3]. In addition, most existing models are specific to individual cancers, with only one risk prediction model to our knowledge developed to predict the risk of multiple cancers combined [21]. However, several risk factors (e.g., smoking) are common in many cancer types [21,29]. Most previous models also include risk factors that are difficult to measure or not readily available (e.g., biomarkers) [23,24,25,26,27,28]. Given that general practitioners often serve as the first point of contact for cancer symptom assessment [30], prediction models that include readily accessible sociodemographic and health-related predictors of cancer stage at diagnosis may help reduce delays in diagnosis and enhance the identification of individuals at risk for late-stage diagnosis.

This feasibility study aimed to develop sex-specific partial proportional odds risk prediction models to identify shared risk factors associated with cancer stage at diagnosis for all cancer types using data from a large population-level prospective cohort study in Alberta, Canada. Prediction models were externally validated on a population-level cohort in British Columbia (BC), Canada.

## 2. Materials and Methods

### 2.1. Data Sources

We used data from two prospective cohorts to develop and externally validate risk prediction models for cancer stage at diagnosis. Both cohorts contributed to the Canadian Partnership for Tomorrow’s Health (CanPath), a pan-Canadian prospective cohort that has collected genetic, clinical, behavioral, and environmental data to support research on cancer and other chronic diseases [31].

Models were developed using data from Alberta’s Tomorrow Project (ATP). From 2000 to 2015, 55,000 adults living in Alberta, Canada aged 35–69 years with no reported history of cancer (excluding non-melanoma skin cancer) were enrolled in ATP [32,33]. The current study used data from ATP participants enrolled between 2001 and 2009 who completed the Health and Lifestyle Questionnaire (HLQ) and consented to have their data linked to administrative databases (>99%) (*n* = 26,554) [32,33]. The HLQ examined sociodemographic characteristics (e.g., household income) and health-related factors (e.g., family history of cancer) [34]. ATP participants also completed the Canadian Diet History Questionnaire [35] and the Past Year Total Physical Activity Questionnaire [36]; however, data from these questionnaires were not used in the current study because of differences between data collection methods used in ATP and the BC Generations Project (BCGP).

External validation was performed using data from the BCGP. From 2009 to 2016, nearly 30,000 adults between the ages of 35 and 69 years living in BC, Canada, were enrolled in the BCGP [37]. Data were collected using the Health and Lifestyle Core Questionnaire (HLCQ) [37]. The HLCQ assessed sociodemographic characteristics and health-related factors similar to the HLQ used in ATP, in addition to diet history and physical activity [38].

### 2.2. Outcome

This study included all incident cancers (excluding non-melanoma skin cancer) diagnosed from the year of cohort enrollment to January 2018 (ATP) and January 2021 (BCGP). Due to the small number of some cancer types, all cancers were combined to identify shared risk factors associated with cancer stage at diagnosis. Cancer stage at diagnosis was defined according to the Tumor, Node, Metastasis staging system, whereby cancers are staged from stage I (cancer is small and contained) to IV (cancer has metastasized) [3]. Cancer type and stage at diagnosis among ATP and BCGP participants were obtained through linkage with the Alberta Cancer Registry [32] and the BC Cancer Registry [37], respectively. Cancer stage at diagnosis among males was categorized as stages I to IV. Among females, cancer stages III and IV were combined because of their low frequencies (i.e., stage I, stage II, and stage III/IV). Participants with missing cancer stage information were excluded.

### 2.3. Predictors

Candidate predictors associated with cancer stage at diagnosis were selected based on evidence from the literature and were subsequently limited to those that were available in both ATP and BCGP cohorts (Appendix A). Due to the unavailability of data or differences in measurement tools, we were unable to include data on dietary intake, physical activity, or physical measurements (e.g., weight). Missing predictor values were imputed using mean value replacement under a missing at random assumption for continuous variables (age at diagnosis, number of pregnancies), and were replaced by the reference group for categorical variables. Continuous variables were evaluated for non-linear relationships. Age at diagnosis was scaled by 10 to improve interpretability. A total of 22 and 29 sociodemographic and health-related variables were considered for males and females, respectively.

### 2.4. Model Development

Preliminary analyses have been described elsewhere [7,8,9] but briefly included centering of continuous predictors, removal of binary predictors with low frequencies (i.e., <5), and collapsing of levels of categorical variables with low frequencies.

Associations between predictor variables and cancer stage at diagnosis among ATP participants were modelled using partial-proportional odds (PPO) models. Owing to the presence of sex-specific cancer types, separate analyses were conducted for males and females. Proportional and partial-proportional univariable screening was performed to assess the association of each variable with cancer stage. Variables that passed univariable screening (*p* < 0.15) were subsequently considered for stepwise variable selection. Stepwise selection was adopted, with the proportionality assumption and interactions assessed at each step. A significance level of *p* < 0.05 was used to determine which variables remained in the final model. The influence of outliers was investigated by computing the differences in effect and fit (DFBETAS and DFFITS, respectively) per participant. Model checking included goodness-of-fit tests, indices of model performance, and diagnostics for binary logistic regression per dichotomization of stage at diagnosis owing to the shortage of available PPO model checking methods. All analyses were performed using R (version 4.1.1, R Foundation for Statistical Computing, Vienna, Austria) and SAS software (version 9.4, SAS Institute Inc., Cary, NC, USA).

### 2.5. Model Performance and External Validation

Discrimination (ability to distinguish between outcomes) and calibration (accuracy of risk predictions) of the ATP-derived PPO models were evaluated among male and female BCGP participants. Discrimination was assessed using the ordinal concordance statistic (ORC) [39]. The ORC is interpreted as the probability that the model can correctly rank observations belonging to two different outcome categories and is the average of all pairwise C-statistics [39]. Calibration levels were assessed using calibration intercepts and slopes, the estimated calibration index (ECI), and graphically using flexible calibration plots [40]. To examine calibration of ordinal prediction models, dichotomizations of cancer stage were created (e.g., for males, stage I vs. stages II to IV, stages I and II vs. stages III and IV, and stages I to III vs. stage IV) [40]. A calibration intercept of zero and calibration slope of 1 indicate that the model does not over- or under-predict risk and is not over- or under-fit to the derivation data, respectively [41]. Flexible calibration curves per dichotomization of cancer stage were generated by plotting the predicted risk against the derived observed risk with curves lying close to the diagonal line, indicating good calibration [42]. The ECI measures the deviation of the flexible calibration plots from the diagonal and should be close to zero [42].

## 3. Results

In total, 2227 and 1845 incident cancers were identified in the ATP and BCGP cohorts, respectively. We excluded 555 ATP and 990 BCGP participants with missing cancer staging data (Table 1). Other cancers aggregated by site for each cohort are presented in Appendix A. Missing predictor data were minimal (<5%) but imputed using our described strategy. Non-linearity was not detected in the continuous predictors.

### 3.1. Model Development

#### 3.1.1. ATP Cohort Characteristics

Our study included 2112 participants (903 males and 1209 females) who developed cancer until January 2018 (Figure 1). Among males, 21.7%, 40.9%, 16.6%, and 20.8% of cancers were diagnosed at stages I, II, III, and IV, respectively, with an average age at diagnosis of 64 years (Appendix A). Among females, 43.1%, 25.7%, and 31.2% of cancers were diagnosed at stage I, II, and III/IV, respectively, with an average age at diagnosis of 62 years. Most participants were Caucasian (96.9% male and 97.3% female) and employed full or part-time (66.8% male, 59.7% female). Few participants reported being former or current smokers (21.6% males and 24.0% females). Most participants had a family history of cancer (57.0% male, 58.1% female). Almost all (90.2%) females had a history of pregnancy, and 24.6% and 8.4% had hysterectomy or oophorectomy, respectively. In terms of cancer screening, 85.4% and 94.4% of females had previously undergone a mammogram or Papanicolaou smear test, respectively, whereas 43.1% of males had undergone a prostate-specific antigen (PSA) blood test. Fewer participants had ever undergone a blood stool test (37.3% males, 37.7% females) or a sigmoidoscopy or colonoscopy (23.4% males and 26.6% females).

#### 3.1.2. Predictors of Cancer Stage at Diagnosis in the ATP Cohort

The results of the final sex-specific PPO models are summarized in Table 2 and Table 3. Among males, older age was associated with an earlier stage at diagnosis (odds ratio (OR) 0.84, 95% confidence interval (CI) 0.72, 0.99 for every 10 years additional age). Full- or part-time current employment status was associated with an earlier stage at diagnosis when comparing stage IV with stages I to III and when comparing stages III and IV with stages I and II (OR 0.49; 95% CI 0.34, 0.70, and OR 0.72; 95% CI 0.53, 0.99, respectively). PSA blood testing was associated with an earlier stage at diagnosis when considering stage IV or stages III and IV as late-stage (OR 0.68; 95% CI 0.48, 0.96, and OR 0.69; 95% CI 0.51, 0.92, respectively). Former or current smoking was associated with a higher risk of late-stage diagnosis (OR 2.34; 95% CI 1.66, 3.29 and OR 1.44; 95% CI 1.10, 1.90, respectively). A family history of heart attack trended towards a greater risk of later-stage diagnosis, although this did not reach statistical significance (OR 1.26; 95% CI 0.98, 1.63).

Among females, having a mammogram or a sigmoidoscopy or colonoscopy was associated with an earlier stage at diagnosis (OR 0.69; 95% CI 0.50, 0.95 and OR 0.74; 95% CI 0.58, 0.95, respectively). Older age was associated with a later stage at diagnosis when comparing stages III and IV with stages I and II (OR 1.40; 95% CI 1.21, 1.61 for every 10 years additional age). Reproductive health factors, including a greater number of pregnancies and history of hysterectomy, were associated with a later-stage diagnosis (OR 1.08; 95% CI 1.01, 1.14 and OR 1.32; 95% CI 1.03, 1.70, respectively).

### 3.2. External Validation

#### 3.2.1. BCGP Cohort Characteristics

A total of 855 (298 males and 557 females) BCGP participants who developed cancer between 2009 and 2021 were included in the study (Figure 1). Differences in the frequencies and distributions of the predictor variables between ATP and BCGP cohorts are presented in Appendix A. Compared to ATP participants, BCGP participants were slightly older at enrollment and at cancer diagnosis, with an average age at diagnosis of 62 years for females and 66 years for males. There were between-cohort differences in educational level, employment status, total annual household income, and ethnicity (females only). In addition, both male and female BCGP participants had higher proportions of cancer screening and lower smoking prevalence than ATP participants. BCGP female participants were similar to ATP female participants in terms of history of menopausal hormone replacement therapy use and having had a hysterectomy. Fewer female BCGP participants had a history of pregnancy compared to those in ATP, while more female BCGP participants reported undergoing oophorectomy than female ATP participants.

#### 3.2.2. Model Performance in the BCGP Cohort

The performance of the risk prediction models differed according to sex (Table 4 and Figure 2). When applied to BCGP participants, females had a slightly lower ORC of 0.53 compared to males (ORC of 0.58). The calibration intercepts and slopes for males were 0.11 (95% CI −0.17, 0.37) and 0.67 (95% CI −0.30, 1.69) for stage I vs. stages II to IV; −0.19 (95% CI −0.42, 0.05) and 1.43 (95% CI 0.68, 2.23) for stage I and II vs. stages III and IV; and 0.08 (95% CI −0.21, 0.39) and 0.86 (95% CI 0.15, 1.59) for stages I to III vs. stage IV, respectively. The calibration intercepts and slopes for females were 0.06 (95% CI −0.11, 0.22) and 0.62 (95% CI −0.08, 1.33) for stage I vs. stages II to III/IV; and −0.04 (95% CI −0.22, 0.15) and 0.53 (95% CI −0.01, 1.08) for stage I and II vs. stages III/IV, respectively. The female flexible calibration plot showed better agreement between the prediction and reference line than the male flexible calibration plot (Figure 2). The corresponding ECIs for females and males were 0.32 and 1.11, respectively, indicating that the risk predictions for females were more moderately calibrated than those for males.

## 4. Discussion

In this feasibility study, we developed and externally validated sex-specific PPO models that identified sociodemographic and health-related factors associated with cancer stage at diagnosis. Prediction models for both males and females showed low discriminative performance in distinguishing between stages at diagnosis. Calibration was better in males than in females.

Our risk prediction models analyzed data from male and female ATP participants. Among males, former or current smoking was associated with late-stage cancer diagnosis, while older age at diagnosis, full- or part-time employment, and history of PSA testing were associated with earlier cancer stage at diagnosis. Among females, a history of mammography, sigmoidoscopy, or colonoscopy was associated with earlier-stage diagnoses, whereas older age at diagnosis, a greater number of pregnancies, and a history of hysterectomy were associated with late-stage cancer diagnosis. Our findings are consistent with some, but not all, previous studies. For instance, smoking has previously been identified as a risk factor for advanced-stage cancer in some studies [10,13], with no associations found in others [7]. Several studies have found associations between younger age (e.g., <50 years) and later stage at diagnosis [13,14], while one study found that this association was cancer-specific, with older age associated with advanced breast cancer among females but earlier lung cancer stage at diagnosis among both males and females [43]. In addition, cancer screening participation has been associated with early-stage diagnoses in most studies [7,9,44]. Finally, previous studies have found associations between hysterectomy and later-stage cancer diagnoses [45]; however, this association may also depend on cancer type. For instance, in a previously published study involving ATP participants, we found that hysterectomy was associated with a lower risk of late-stage colorectal cancer [8]. Given the differences in study designs, population groups, sample sizes, and cancer types, findings across studies cannot be directly compared. In addition, our study grouped all cancers and examined associations among males and females separately, whereas most previous studies focused on individual cancers and combined males and females in their analysis. 

We externally validated our prediction models by examining model discrimination and calibration (mean and weak). Our models showed poor discrimination, with ORCs for males and females near 0.5, indicating that the model’s ability to separate outcomes in the validation data was similar to that of a random model [39]. The calibration intercepts for each dichotomization of cancer stage at diagnosis for both male and female models were near zero, and the corresponding 95% CIs included zero, indicating that there was little difference between the observed and average predicted proportions of each category of cancer stage at diagnosis among BCGP participants (mean calibration) [40,41]. When examining calibration slopes, the models for males performed better than those for females. Notably, although both male and female models were slightly above or below the target calibration slope of 1, the corresponding 95% CIs contained 1, indicating that the models were not over- or under-fit to ATP data (weak calibration) [40,41]. However, for all outcome dichotomizations of the female models and one outcome dichotomization of the male models (i.e., stage I vs. stages II–IV), the 95% CIs also contained 0, suggesting that the predicted stage at diagnosis was not associated with the observed stage at diagnosis of BCGP participants for those dichotomizations. Conversely, the flexible calibration plots and corresponding ECI showed poorer predictions for males relative to females. The flexible calibration plot for males often deviated from the diagonal for each stage and had an ECI near 1 (random model), indicating inadequate moderate calibration. The flexible calibration plot for females for each stage ran closer to the diagonal and with an ECI closer to 0 than to 1, indicating more moderately calibrated predictions. However, there is no reliable test for moderate calibration [41] so only visual interpretations are used. These calibration results are likely due to factors such as the smaller validation sample size of BCGP males and the collapsing of stages III and IV in the models for females [17]. Distributional differences between cohorts for most predictors included in the final models (Appendix A) could partly explain the low ORCs and the lack of moderately calibrated male risk predictions.

The development of ordinal risk prediction models for cancer outcomes is novel and, to our knowledge, only one has been previously developed to predict cancer stage at diagnosis [23,24]. The Assessment of Different Neoplasias in the Adnexa (ADNEX) model was developed to characterize ovarian tumors as benign, borderline, stage I, stage II-IV, and secondary metastatic ovarian cancer [23,24,25,26,27]. The ADNEX model includes clinical and ultrasound predictors and has been externally validated in various population groups [24,25,26,27,28]. The model showed good discrimination with a polytomous discrimination index ranging from 0.56 to 0.58 (with a possible score ranging from 0.2 (poor) to 1.0 (perfect)) [24,26]. The ADNEX also showed good discriminative performance when each outcome category was compared to a reference (i.e., benign tumor), with an area under the receiver operating characteristic curve (AUC) ranging from 0.72 to 0.99 [24,25]. However, the model poorly distinguished between borderline and stage I ovarian tumors (AUC 0.54) [25]. Calibration of the ADNEX was also good based on the calibration plots [24,26]. Although the predictive performance of the ADNEX is higher than that of our risk prediction models, it is difficult to compare the models given that the ADNEX model is specific to ovarian tumors and was developed to primarily examine predictors obtained from ultrasound measurements [23,24].

### 4.1. Strengths

Our prediction models are the first to our knowledge to apply cancer stage at diagnosis as an ordinal outcome for all cancers using novel statistical methods [39,40,42]. Maintaining the ordinal nature of stage at diagnosis, rather than dichotomizing our outcome, helped minimize information loss and considered the extent of the disease, which may better inform decision making. We used PPO models, which relax the proportional odds assumption, and thus allow more flexibility with regard to variable inclusion in our models. We externally validated our prediction models in an independent but similar cohort using both calibration and discrimination. Evaluating predictive performance using an external dataset enhances the generalizability of model predictions and is a crucial step prior to its application and use in clinical practice [46]. In addition, systematic reviews have found that while most studies assess discrimination of prediction models, few assess calibration [19,20]. Calibration is particularly important, as it provides an indication of a model’s clinical usefulness [47]. We used calibration intercepts and slopes, rather than the Hosmer–Lemeshow test, which has several weaknesses [41]. Our prediction models were developed and externally validated using two large population-level cohorts in Canada. The prospective cohort study design allowed for the examination of risk factors before cancer diagnosis, thus minimizing recall bias. Cancer cases were identified through cancer registries, minimizing selection and information bias. Using sociodemographic and health-related variables, rather than more complex factors such as genetics, can enhance the reproducibility of the models. Finally, by incorporating risk factors that can be assessed during a routine medical visit, our prediction models can help general practitioners quickly and easily identify individuals at risk for later-stage cancers, thereby enhancing earlier cancer detection, treatment efficacy, and survival [3].

### 4.2. Limitations

This study has some limitations. First, there were insufficient data to explore individual cancer types separately. Validation results for separate PPO models for breast cancer (females), prostate cancer (males), and lung plus colorectal cancer had very poor discrimination and calibration (results available from corresponding author upon request). Thus, all cancers were pooled. Many cancers have distinct risk factors; therefore, creating sex-specific models to predict stage at diagnosis across all cancer types may have reduced our models’ predictive performance. However, most cancers also have shared risk factors [21,29], which our study identified. We demonstrated that it is feasible to develop and externally validate risk prediction models for cancer stage at diagnosis. In addition, while we used a relatively comparable cohort to externally validate our prediction models, there were some baseline differences between cohorts. There were also differences in the distribution of stage at diagnosis across cancer types, particularly among females, and data from the ATP and BCGP cohorts were collected approximately 10 years apart. Differences in cohort baseline characteristics and temporal trends between baseline data collection time points (e.g., cancer screening and treatment policies) may have reduced the predictive performance of our models [17,48]. Second, the predictors included in our models were limited to data that were available and similar in both cohorts. Thus, we were unable to include several important risk factors for cancer stage at diagnosis, such as dietary factors and race/ethnicity [8,9,13]. Nonetheless, we were able to include important predictors of cancer stage at diagnosis such as smoking and screening participation. Third, cancer stage at diagnosis was missing for 20.8% of cancer cases among ATP participants and 53.7% of BCGP participants, which may have introduced selection bias if participants with observed cancer stage at diagnosis were systematically different from those with missing values [49]. These missing data were assumed to be missing at random due to differences in the implementation of stage information being collected by the two provincial cancer registries. The varying proportions of missing data across cancer subtypes could also have impacted the validation results. The most common cancers were the most frequent ones in both cohorts including subtypes in the other cancers category. Thus, the most likely impact would be on statistical power for the validation results and not systematic biases, although this could not be confirmed.

With ongoing data harmonization across cohorts within CanPath [31], our prediction models can be updated with additional relevant predictors, which may increase their predictive power and performance. Data harmonization will also provide opportunities to externally validate our predictive models in other Canadian cohorts to further enhance generalizability [48]. Implementing risk prediction models into clinical practice requires extensive evaluation, including internal and external validation, followed by impact assessment to assess the model’s clinical performance and usefulness in clinical decision-making. Thus, optimizing the performance of our models (e.g., ongoing updating and refinement and further external validation) is a crucial step prior to implementation into clinical practice [50].

## 5. Conclusions

We developed and externally validated sex-specific PPO models to predict factors associated with cancer stage at diagnosis among ATP participants. We identified several sociodemographic and health-related risk factors associated with stage at diagnosis for all cancer types among males and females. External validation of our models showed poor discrimination and meanly and weakly calibrated risk predictions. Updating our models with additional predictors and ongoing external validation in other cohorts may help improve predictive performance, as our study found that it is feasible to do so. Prediction models that identify risk factors associated with cancer stage at diagnosis can help identify individuals at a higher risk of developing late-stage cancer, and thus greatly improve cancer survival. Further research is required to develop more robust prediction models.

## Figures and Tables

**Figure 1 cancers-15-03545-f001:**
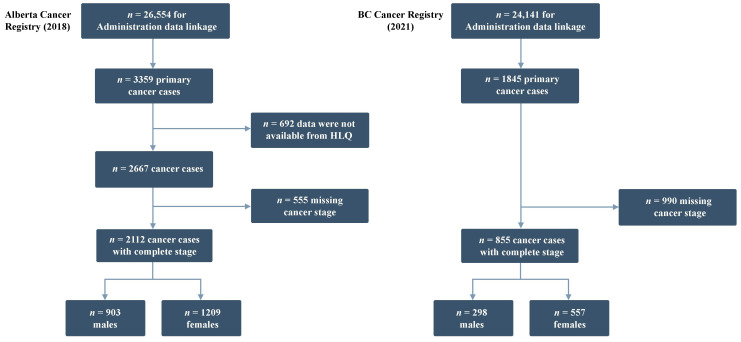
Flow of Alberta’s Tomorrow Project and British Columbia Generations Project participants diagnosed with cancer with available data on cancer stage. BC: British Columbia, HLQ: Health and Lifestyle Questionnaire.

**Figure 2 cancers-15-03545-f002:**
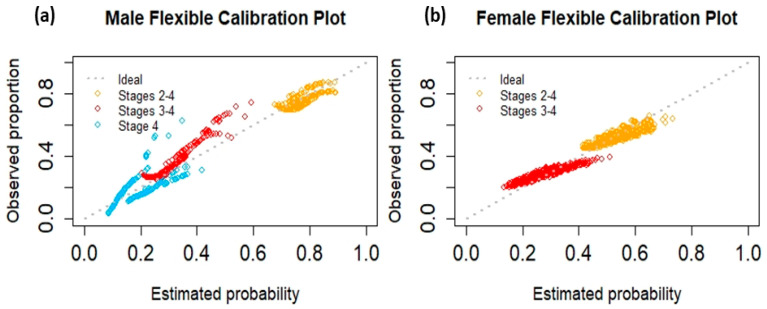
Flexible calibration plots of predicted probabilities for each cancer stage at diagnosis for males (**a**) and females (**b**).

**Table 1 cancers-15-03545-t001:** Frequency of cancer cases and missing staging data by sex for the four most prevalent cancers among Alberta’s Tomorrow Project (*n* = 2112) and British Columbia Generations Project (*n* = 855) participants.

	ATP			BCGP		
	Cancer Cases from the Alberta Cancer Registry (*n*)	Cancer Staging Available (*n*)	Missing Cancer Staging (%)	Cancer Cases from the BC Cancer Registry (*n*)	Cancer Staging Available (*n*)	Missing Cancer Staging (%)
			**Male**			
Breast cancer						
Lung cancer	95	88	7%	27	21	22%
Colorectal cancer	135	128	5%	36	24	33%
Prostate cancer	461	410	11%	228	171	25%
Other cancers	484	277	43%	444	82	82%
Total	1175	903	23%	735	298	59%
			**Female**			
Breast cancer	530	526	1%	358	293	18%
Lung cancer	149	133	11%	61	47	23%
Colorectal cancer	144	139	3%	61	47	23%
Prostate cancer						
Other cancers	669	411	39%	630	170	73%
Total	1492	1209	19%	1110	557	50%

ATP, Alberta’s Tomorrow Project; BC, British Columbia; BCGP, British Columbia Generations Project.

**Table 2 cancers-15-03545-t002:** Predictors of cancer stage at diagnosis among male Alberta’s Tomorrow Project participants using Partial Proportional Odds models (*n* = 903).

	Cancer Stage (PPO)	OR (95% CI)	*p* Value
Age at diagnosis (10 years)		0.84 (0.72, 0.99)	0.04
Employed full- or part-time			
Yes vs. No	IV vs. I and II and III	0.49 (0.34, 0.70)	<0.001
	III and IV vs. I and II	0.72 (0.53, 0.99)	0.04
	II and III and IV vs. I	0.77 (0.53, 1.10)	0.15
Heart attack family history			
Yes vs. No		1.26 (0.98, 1.63)	0.07
History of PSA blood test			
Yes vs. No	IV vs. I and II and III	0.68 (0.48, 0.96)	0.03
	III and IV vs. I and II	0.69 (0.51, 0.92)	0.01
	II and III and IV vs. I	1.01 (0.73, 1.40)	0.96
Smoking status			
Never smoker		1.0	
Past smoker		1.44 (1.10, 1.90)	0.01
Current smoker		2.34 (1.66, 3.29)	<0.001

Note: An entry in the cancer stage column indicates that the covariate has different effects on different cancer stages (i.e., does not meet the PO assumption); otherwise, the covariate meets the PO assumption, where the coefficients are equal for IV vs. I and II and III, II and IV vs. I and II, and II and III and IV vs. I. OR, odds ratio; PPO, partial proportional odds; PSA, prostate-specific antigen.

**Table 3 cancers-15-03545-t003:** Predictors of cancer stage at diagnosis among female Alberta’s Tomorrow Project participants using Partial Proportional Odds models (*n* = 1209).

	Cancer Stage (PPO)	ORs (95% CI)	*p* Value
Age at diagnosis (10 years)	III/IV vs. I and II	1.40 (1.21, 1.61)	<0.001
	II and III/IV vs. I	1.09 (0.96, 1.25)	0.19
Number of pregnancies		1.08 (1.01, 1.14)	0.02
History of sigmoidoscopy or colonoscopy			
Yes vs. No		0.74 (0.58, 0.95)	0.02
History of mammogram			
Yes vs. No		0.69 (0.50, 0.95)	0.02
History of hysterectomy			
Yes vs. No		1.32 (1.03, 1.70)	0.03

Note: An entry in the cancer stage column indicates that the covariate has different effects on different cancer stages (i.e., does not meet the PO assumption); otherwise, the covariate meets the PO assumption, where the coefficients are equal for IV vs. I and II and III, II and IV vs. I and II, and II and III and IV vs. I. OR, odds ratio; PPO, partial-proportional odds.

**Table 4 cancers-15-03545-t004:** Validation results for the British Columbia Generations Project data sets.

	Calibration Intercepts (95% CI) per Outcome Dichotomy				
**Model**	Y≤1	Y≤2	Y≤3	**ECI**	**ORC**
Male PPO	0.11 (−0.17, 0.37)	−0.19 (−0.42, 0.05)	0.08 (−0.21, 0.39)	1.11	0.58
Female PPO	0.06 (−0.11, 0.22)	−0.04 (−0.22, 0.15)		0.32	0.53
	**Calibration slopes (95% CI) Per outcome dichotomy**				
	Y≤1	Y≤2	Y≤3		
Male PPO	0.67 (−0.30, 1.69)	1.43 (0.68, 2.23)	0.86 (0.15, 1.59)		
Female PPO	0.62 (−0.08, 1.33)	0.53 (−0.01, 1.08)			

ECI, estimated calibration index; ORC, ordinal C statistic; PPO, partial proportional odds. Note: Male Y ≤ 1 = stage I vs. stages II to IV, Y ≤ 2 = stages I & II vs. stages III & IV, and Y ≤ 3 = stages I to III vs. stage IV; female Y ≤ 1 = stage I vs. stage II & III/IV and Y ≤ 2 = stages I & II vs. stages III/IV.

## Data Availability

Access to individual-level data is available in accordance with the Health Information Act of Alberta and Alberta’s Tomorrow Project (ATP) Access Guidelines at https://myatpresearch.ca/DataAccess (accessed on 11 January 2018). The data used for external validation are available from the BCGP (https://www.bcgenerationsproject.ca (accessed on 10 September 2020)).

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
