# Peer review of "Development and External Validation of Partial Proportional Odds Risk Prediction Models for Cancer Stage at Diagnosis among Males and Females in Canada"

_cancers, 2023, doi:10.3390/cancers15143545_

Round 1
Reviewer 1 Report
The authors developed and validated a prediction model for the disease stage. I would like to thank the authors for this work.
The manuscript is really well written and conducted study. Some comments need to be addressed.
# Introduction
1. Cancer state at diagnosis is inevitable related to screening as the authors already mentioned.
- Comment: do the authors think that the region of Alberta is a general sample of the current screening practices among Canadians? If this is not the case, it might be an explanation of the poor validation.
# Material and methods
1. Comment. The authors excluded participants with missing cancer stage. In case of missing covariates, did the authors perform imputation methods?
2. Comment. The authors perform a partial proportional odds model. Was this decision based on previous results? If not, which variable violated the proportion odds assumption?
3. Comment. Did the authors consider fitting non-linear terms for continuous variables?
# Results
1. Table 1. The group of "other cancers" is substantial. Can the authors define this group better in supplemental material (only those cancers which accounts for more than 10%?)
2. Comment. The authors could consider rescaling the variable "age at diagnosis" by 10 years to get a more interpretable coefficient.
Reviewer 2 Report
Thank you asking me to review. I think this is an excellent study conducted by an expert group and making very good use of two different population sources of health data in Alberta. I think the concept for the study is very important. The paper is very well presented and the methods transparent and reproducible. Biggest limitation which the authors acknowledge is the lack of ability (sample size) to specify the work by cancer site. Breast and lung cancer will have different predictors and prognosis even for the same stage. I have the following comments for the authors to consider.
1. The statistical section is very thorough, but left me with a small number of comments and questions, which I am sure the authors can clarify:
a. How was age handled, was it as a continuous variable? Previous research shows that the effects are not entirely continuous across the age spectrum. Using splines, for example, would have allowed for a better appreciation of age associations. See also https://pubmed.ncbi.nlm.nih.gov/33564123/ which I think is a paper that can be cited in any case. Can the authors please comment.
b. I appreciate that the ORC is Ordinal C Statistic'. Is this equivalent to standard C statistic? Please explain. Can area-under-the curve and ROC curves be used in addition to communicate the findings?
c. The authors indicate that calibration was poor, but from the visuals, the non-expert reader gets a different impression. Can the authors please expand a bit on this section.
2. At the conceptual level, is there value in considering prediction cancer incidence itself, before predicting stage at diagnosis. Please see this paper, which I think should be cited https://pubmed.ncbi.nlm.nih.gov/35954486/
3. Also at the conceptual level, one interpretation of the findings is that it is possible to predict cancer incidence, but not cancer stage, or that conditional on developing cancer, the risk of stage at diagnosis is nearly random. Could the authors consider this possible interpretation and comment if correct or not (in their paper). Would it be worth trying to predict cancer incidence (no matter the stage) in addition. Apologies if this is already done by the authors and missed it in references. It will be worth emphasizing the point as applicable.
On a similar vein is is worth considering different presenting symptoms, if the cancer was not screening detected.
4. Please comment further on possible clinical practice implications. What action could perhaps be supported by the findings, or what other evidence may be needed for translation into practice?
Reviewer 3 Report
Dear Author(s),
Your manuscript entitled “Development and external validation of partial proportional odds risk prediction models for cancer stage at diagnosis among males and females in Canada” provides evidence to support clinical decision-making addressed to cancer patients.
The identification of stage determinants is clinically relevant as the stage at diagnosis determines cancer treatment options and ultimately patient survival. While this is a valuable contribution, the paper presents some methodological weaknesses that need to be addressed.
The main methodological issue concerns the classification of the stage at diagnosis, criteria for classifying stage differ from one cancer type to another, thus compromising the consistency and comparability of the stage variable across different cancer types. This issue could be easily addressed by analyzing data separately by cancer type, but according to your statement in the Materials and Methods section (page 3, line 19 from the top): “Due to small numbers of some cancer types, all cancers were combined …” and in the discussion (limitation section, page 9, 11 line from the bottom): “there were insufficient data to explore individual cancer types separately; thus, all cancers were pooled”, this was not possible.
I wonder if you tried to pool together cancer types which have more consistent classification of stage at diagnosis and/or to pool together men and women data for more frequent cancer types, as is the case for colorectal and lung cancer, thus having enough numbers to run the model and homogeneous data with respect to the outcome variable.
I suggest to run additional analyses for colon-rectum and/or lung cancer men and women combined and for breast cancer among women and prostate cancer among men. In any case, you should address this issue in the discussion, in the section of the limitations of the study.
A second main issue is related to the % of missing stages, which varies from 20.8% on average for ATP study cohort to 53.7% on average BCGP cohort used for validation. This limitation is clearly stated in the discussion (limitation section, page 10, 12 line from the top), however, what is not clearly addressed in the paper is the variability of the % of missing stages across different cancer types, which in turn might have different impact on results.
This issue should be mentioned and critically addressed in the discussion section.
Minor points:
- - Table 1 presenting the study cohort and the cohort used for validation purposes should be split by gender consistently by all analyses and results;
- - Figure 2 is not clear, probably because it is an image, stage categories used for males and females are different, this makes it difficult to compare results by gender. I suggest: to improve the quality of the figure; to uniform stage categories in the two plots; to add a) and b) in Figure 2 to identify male and female plots;
Round 2
Reviewer 1 Report
I would like to thank the authors for revising the manuscript and for addressing my comments.
Regarding section Results, comment 2 (rescaling the variable age at diagnosis per 10 years), I would like to suggest to update the table in the manuscript.
Author Response
Thank you for your positive comments and the suggestion to :
Regarding section Results, comment 2 (rescaling the variable age at
diagnosis per 10 years), I would like to suggest to update the table in
the manuscript.
We have revised the Methods section to include this statement (highlighted in yellow as well as tracked changes):
Age at diagnosis was scaled by 10 to improve interpretability.
In the Results section, we revised tables 2 and 3 with age at diagnosis scaled by 10. The corresponding Odds Ratios, 95% confidence intervals and p-values associated with these revised tables were also changed and highlighted in yellow to more easily identify the changes.